# PAD Inhibitors as a Potential Treatment for SARS-CoV-2 Immunothrombosis

**DOI:** 10.3390/biomedicines9121867

**Published:** 2021-12-09

**Authors:** Willie Elliott, Maheedhara R. Guda, Swapna Asuthkar, Narasaraju Teluguakula, Durbaka V. R. Prasad, Andrew J. Tsung, Kiran K. Velpula

**Affiliations:** 1Department of Cancer Biology and Pharmacology, University of Illinois College of Medicine at Peoria, Peoria, IL 61605, USA; wjellio2@uic.edu (W.E.J.); gmreddy@uic.edu (M.R.G.); asuthkar@uic.edu (S.A.); Andrew.J.Tsung@ini.org (A.J.T.); 2VTD Biopharma, Bangalore 560105, India; narasaraju0425@gmail.com; 3Department of Microbiology, Yogi Vemana University, Kadapa 516003, India; durbaka@gmail.com; 4Department of Neurosurgery, University of Illinois College of Medicine at Peoria, Peoria, IL 61605, USA; 5Illinois Neurological Institute, Peoria, IL 61603, USA; 6Department of Pediatrics, University of Illinois College of Medicine at Peoria, Peoria, IL 61605, USA

**Keywords:** SARS-CoV2, COVID-19, PAD, NET

## Abstract

Since the discovery of severe acute respiratory syndrome coronavirus 2 (SARS-CoV-2) in December 2019, the virus’s dynamicity has resulted in the evolution of various variants, including the delta variant and the more novel mu variant. With a multitude of mutant strains posing as challenges to vaccine efficacy, it is critical that researchers embrace the development of pharmacotherapeutics specific to SARS-CoV-2 pathophysiology. Neutrophil extracellular traps and their constituents, including citrullinated histones, display a linear connection with thrombotic manifestations in COVID-19 patients. Peptidylarginine deiminases (PADs) are a group of enzymes involved in the modification of histone arginine residues by citrullination, allowing for the formation of NETs. PAD inhibitors, specifically PAD-4 inhibitors, offer extensive pharmacotherapeutic potential across a broad range of inflammatory diseases such as COVID-19, through mediating NETs formation. Although numerous PAD-4 inhibitors exist, current literature has not explored the depth of utilizing these inhibitors clinically to treat thrombotic complications in COVID-19 patients. This review article offers the clinical significance of PAD-4 inhibitors in reducing thrombotic complications across various inflammatory disorders like COVID-19 and suggests that these inhibitors may be valuable in treating the origin of SARS-CoV-2 immunothrombosis.

## 1. Introduction

As of 13 August 2021, the World Health Organization (WHO) reported a global 205,338,159 confirmed number of COVID-19 cases, with 4,333,094 (2.11%) of these confirmed cases resulting in mortality (https://COVID19.who.int, accessed on 28 October 2021). According to Johns Hopkins Coronavirus Research Center, which tracks the rate of COVID-19 cases over 28 days, as of 14 August 2021, the United States, India, Indonesia, Brazil, and the United Kingdom have been leading in COVID-19 cases with reports well above one million over the last 28 days (https://coronavirus.jhu.edu/map.html, accessed on 28 October 2021). Importantly, the SARS-CoV-2 that the world encountered starting on 20 December 2019, is not the sole strain of SARS-CoV-2 that is causing illness and death. As of 6 July 2021, there has been numerous variants of SARS-CoV-2 that have begun to take public health interest, including the alpha variant (B.1.1.7), beta variant (B.1.351), Gamma variant (P.1 variant), Cal.20C variant, Mu (B.1.621) [1], and most notably the delta variant (B.1.617.2) [2]. The Delta variant, first discovered in Maharashtra, India, gained attention from its spread amongst the United Kingdom and relatively recent spread to the Midwest of the United States [2,3]. The variant, harboring multiple mutations in the spike protein, gives it marked resistance compared to other strains of SARS-CoV-2, such as the alpha strain [3]. Mutations of the spike protein decrease the efficacy of many of the current vaccine options available and offer a continuing challenge in SARS-CoV-2 pharmacotherapy as we continue to treat current variants and potential novel variants. Thus, the importance of developing novel treatments specific to SARS-CoV-2 and related complications, such as thrombosis, remains critical in the future of treatment. Immunothrombosis, immune-related thrombotic complications, remains a pressing predicament for COVID-19 patients with mild and severe symptoms, often resulting in in-patient use of thromboprophylaxis to prevent the risk of venous thromboembolism [4]. However, such prophylactic intervention often involves targeting anti-coagulation factors to assist in coagulation factor breakdown or coagulation factor inhibitors, thereby leading to an approach of treating a product of the cause rather than the origin, especially inflammatory components such as neutrophil extracellular traps (NETs) and NET components, such as citrullinated histones as being the constituents involved in vascular occlusion [5]. Use of peptidylarginine deiminase (PAD) inhibitors, specifically the PAD-4 isozyme, offers valuable insight into controlling products of inflammation that drive immunothrombosis in SARS-CoV-2.

## 2. Netosis and Citrullinated Histones Potentiate Immunothrombosis

The prevalence of thrombotic complications has recently begun to surface in patients hospitalized with COVID-19, leading to devastating outcomes, such as cerebrovascular events [6,7,8]. Although novel manifestations, such as strokes, are becoming more apparent in individuals infected with SARS-CoV2, the mechanism behind thrombosis remains unclear [9]. NETs have been demonstrated to play a critical role in mediating thrombosis in blood vessels [10,11], which has been attributed to the thrombogenic nature of histone proteins [11]. Considering the pathologic implications of NETs, mechanistically degrading or inhibiting NETs formation is essential in preventing immunothrombosis [12,13]. The question as to how a host degrades NETs in vivo is ambiguous. One study sought to provide a solution to the question by observing the impact on DNase1 and DNase1L3 knockout mice and the resultant NET formation and thrombogenesis that occurred. To first test that these enzymes degrade NETs, the researchers subjected the knockout mice to NETs generated in vitro and compared the observations to wild-type (WT) mice expressing both enzymes. Results demonstrated that the wild-type mice were able to degrade NETs, while the knockout mice could not. These findings remained consistent in vivo, where researchers took the knock-out mice and WT mice and administered granulocyte colony-stimulating factor (G-CSF), an agent that triggers neutrophilia; mice not expressing the DNase enzymes expressed a high neutrophil count in serum and organs, while WT mice grew normal and relatively healthy. Similar results are captured when DNase1 is instead infused in deep vein thrombosis (DVT)-induced mice, showing a significant reduction in the mice that developed DVT after induction [14]. These findings demonstrate that DNase1 and DNase1L3 are essential enzymes in degrading NETs, a notable consideration when accounting for the constituents that makeup NET complexes. Interestingly, Alcazar et al. also found that knockout mice with G-CSF induced chronic neutrophilia displayed intravascular hematoxylin-positive clots that fully or partially occluded blood vessels in lungs, liver, and kidneys. Expression of DNase1 and DNase1L3 in WT mice was sufficient to prevent the vascular occlusions, suggesting therapeutic implications of DNases in preventing NETs-mediated pathology.

The constituents of the clots arising in select COVID-19 patients are classic in comparison to existing literature, demonstrating an infiltration of NETs, citrullinated histones, and von Willebrand factor (vWF) [14,15]. Imaging of thrombi in surgically DVT-induced mice illustrated loci positively stained for Gr-1, an antigen present on polymorphonuclear leukocytes, and citrullinated histone 3 (CitH3). Immunostaining of neutrophils in a separate study displayed analogous findings of CitH3 in WT mice exposed to Klebsiella pneumoniae but intriguingly demonstrated an absence of CitH3 in PAD-4 negative neutrophils [14]. The presence of CitH3 suggests a potential release of extracellular decondensed chromatin, as seen in NETs, which is associated with clot formation (Figure 1). In fact, when PAD deficient mice are subjected to surgically induced DVT, they exhibited a significant reduction in clot formation. Notably, upregulation of PAD isozymes in SARS-CoV-2 patients is observed in various forms of the protein. Visualization of RNA by in situ hybridization and further assessment by immunohistochemistry shows elevation of PADI2 and PADI4 in the lungs of five SARS-CoV-2 infected patients when compared to control subjects [16]. Some isozymes of PAD, including PADI1 and PADI6, were elevated but to a less significant degree. Inspection of nucleosomes levels in SARS-CoV-2 infected patients [17] further corresponds with the elevations observed in DVT-induced mice. Patients diagnosed with COVID-19 were observed to have elevated levels of histone variant H3.1 and H3R8 compared to controls, more prominently among patients admitted into intensive care units (ICU) compared to non-ICU patients positive for COVID-19 [17]. The increased NETs and associated turnover products described in prior studies are evidenced to assist in occlusion of vasculature, notably pulmonary vessels [5]. Plasma analysis of COVID-19 patients demonstrated high levels of CitH3, that correlated with elevated leukocyte and granulocyte count. A deeper look within lung tissue samples from patients with COVID-19 found that NETs occlude pulmonary vessels [5,18,19]. Furthermore, NET aggregation displayed dynamic morphology, as the pulmonary vessel clogs were either formed by DNA co-localizing with neutrophil elastase or aggregates of cells containing citrullinated histones [18,19]. Mechanistically, the platelets forming the core of the clot constituents are shown to have marked adhesion to neutrophils when isolated from patients with severe COVID-19 infections [19]. Upon isolation of activated platelet-rich plasma from patients with severe COVID-19 infections, associated high NETs levels corresponded to the platelet-neutrophil adhesion, demonstrating that NETs could be acting as a key mediator for the formation of immunothrombosis in patients infected with the virus [20].

Aside from SARS-CoV2, other microbial infections have been correlated with the formation of NETs and citrullinated histones, as seen in a study examining the response of NETs in mice infected with influenza [21]. Platelets isolated from mice with influenza and exposed to neutrophils from uninfected mice were observed to have a substantially increased NETs level in comparison to platelets from control mice. In a study examining septic patients, increases in CitH3 were observed to be significantly elevated in patients with septic shock compared with those who had non-infectious shock [22]. Mice infected with a lethal dose of influenza were observed to have microvascular thrombi present in small pulmonary vasculature, in proximity with enhanced citrullinated histones, platelets, and fibrin deposition [23]. Enhanced citrullination apparent in other microbial infections displays that measurement of nucleosome levels for tracking SARS-CoV2 disease progression could be non-specific but still may offer as a notable measure of disease prognosis and likelihood of thrombotic complications. NETs and citrullinated histones play a critical role in both the development and constitution of clots; inhibition of these components, associated with thrombosis, may yield insight on treating SARS-CoV-2 clotting complications.

## 3. Overview of PADs and Significance in SARS-CoV-2

Peptidylarginine deiminases (PADs) are a group of enzymes responsible for post-translational modification of arginine residues by citrullination, lending them helpful to processes such as gene or NET activation [24]. There are five isozymes of PAD-4 in humans: PAD1, PAD2, PAD3, PAD4, and PAD6. The various isozymes display structural similarity but interestingly exhibit a tissue-specific pattern [25]. PADs 1–3 and PAD6 are primarily located in the cytoplasm, while PAD4 has a predilection to immune system cells and cancer cell lines such as colorectal derived liver metastatic carcinoma [26] and breast cancer cells [27]. Although PADs can act on a multitude of different substrates, the most classical substrate remains to be histones. Peptidylarginine deiminase 4 (PAD-4) is an enzymatic protein responsible for histone citrullination, in which specific arginine residues are converted to citrulline residues on histone tails, allowing for neutralization of the positive arginine charge and subsequent decondensation of chromatin [14,28]. This chemical modification is critical in NETs release in SARS-CoV-2 [29], respiratory syncytial virus (RSV) [29], endotoxins from bacteria [30], and cryptococcus neoformans fungi [31]. NETs have been implicated in COVID-19 pathogenesis [20] and the ability for a PAD inhibitor to block the formation of citrullinated histones may offer therapeutic benefit to both the thrombotic and inflammatory effects observed in the viral infection.

PAD inhibitors offer an abundance of therapeutic potential (Figure 1), yet the quantity and selectivity of inhibitors apparently is relatively low. Several weaker and reversible PAD inhibitors exist, including paclitaxel (taxol), minocycline, and streptomycin, possessing half-maximal inhibitory concentration (IC_50_), within the millimolar range [32,33]. IC_50_ values are utilized to measure drug efficacy, specifically demonstrating the amount of drug necessary to inhibit a biological process by half and thereby being representative of drug potency [33,34]. Although the IC_50_ values for minocycline and streptomycin toward PAD inhibition have not yet been elucidated, paclitaxel has an IC_50_ of 5 × 10^3^ µM [35], consistent with prior literature stating that weak reversible PAD inhibitors have millimolar range IC_50_. Furthermore, the inhibitory constant (K_i_) for paclitaxel was determined to be between 4.5 × 10^3^ µM and 1.05 × 10^4^ µM at high substrate concentration, although the literature did not specify isozyme specificity for inhibition [36]. The lack of potency of weak PAD inhibitors in addition to some of the side-effects that may arise, make the use of these inhibitors for SARS-CoV-2 treatment insubstantial. Taxol, generically termed Paclitaxel, is a microtubule stabilizing agent commonly used for cancer therapy [37] with reported side effects of myelosuppression [38], a side-effect inconsistent with SARS-CoV-2 treatment. Propitiously, there are more potent and non-reversible PAD inhibitors that appear to be receiving interest from the scientific community as therapeutic options for different disease models. YW3-56, a potent pan PAD inhibitor represented by an IC_50_ of 1–5 µM [39], has been utilized by Wang and colleagues to partially prevent renal ischemia and reperfusion induced by acute kidney injury [40]; streptonigrin has also exemplified PAD inhibitory activity [41]. Cl-amidine is a covalent, irreversible, and potent pan-PAD inhibitor that has been used in various animal disease models to monitor the effect of decreased citH3 and NET levels on disease state and progression; F-amidine is a similar, yet less potent, haloacetamide compound that differs in molecular composition by a single fluorine atom [24,42]. Cl-amidine and F-amidine IC_50_ values are typically within the lower micromolar range for PAD1—4, with F-amidine having more elevated IC_50_ [35,43]. Second generation variations of Cl and F-amidine have been generated, offering increased potency than their less novel counterparts, yet having not received great scientific interest as therapeutics. Furthermore, Ai et al. demonstrates the pharmacotherapeutic potential of BB-Cl-amidine, a potent pan-PAD inhibitor, in reducing inflammatory and thrombotic markers and complications in mouse models [44]. BB-Cl-amidine is measured to have a half maximal effective concentration (EC_50_) of 0.6 µM/L in contrast to ~200 µM/L. Unlike IC_50_, EC_50_ is utilized as a measure of the amount of drug necessary to induce a response halfway between the baseline and maximum [45]. Use of a relatively novel reversible PAD-4 selective inhibitor, GSK199, has been shown to reduce symptoms and associated histopathology and inflammatory markers in collagen-induced arthritis mouse models [46]. Closely related in molecular structure, GSK484, also a reversible PAD-4 selective inhibitor, has shown therapeutic hope in offering as an alternative treatment for heparin-induced thrombocytopenia [47]. GSK 199 and GSK484 are noted to be exceedingly potent to PAD4 inhibition, displaying IC_50_ values significantly lower than Cl-Amidine [48]; additional literature suggests that these values can vary [49], although said variation appears negligible. GSK199 and GSK484 inhibition is observed to display variability in IC_50_ dependent on the concentration of Ca^2+^. PAD4 inhibition through these enzymes is competitive with Ca^2+^ ions, and the substrates specific to PAD isozymes prefer binding in the absence of Ca^2+^. Finally, Thr-Asp-F-amidine (TDFA), is a PAD-4 selective irreversible inhibitor that has been positively correlated with improved acute lung injury in mice [50]. TDFA potency is marked in PAD4 suppression, albeit PAD1-3 attenuation is relatively low in the micromolar range. A summary of the preceding PAD inhibitors and their potencies toward different PAD isozymes are demonstrated in Table 1 through applicable IC_50_ and K_I_ values. Although many other PAD inhibitors exist, scientific literature has not suggested an apparent therapeutic usage for disease treatment in a clinical apparatus.

## 4. Current Treatment of SARS-CoV-2 Immunothrombosis

As of 8 July 2020, a committee of vascular thrombosis experts, physicians, technologists, and other participants created a general guideline for thrombotic complications in patients with SARS-CoV-2 [4]. A consensus was made of thromboprophylaxis for all hospitalized acutely ill patients by subcutaneous unfractionated heparin or low-molecular-weight heparin (LMW heparin) in all COVID-19 positive patients despite severity, to prevent venous thromboembolism (VTE) [4,51,52]. For the treatment of critically ill hospitalized COVID-19 patients, thromboprophylaxis management is not clear, although prophylactic to intermediate doses of LMWH have been associated with better prognosis [51]. In addition, reports of thrombotic complications in ICU patients with COVID-19 as high as 31% have suggested increasing prophylaxis of factor Xa inhibitors, such as enoxaparin, from 40 mg OD to 40 mg twice daily [53]. For suspected pulmonary embolism (PE) or deep vein thrombosis (DVT), separate entities of VTE, classical approach of judgement through a modified Wells score and subsequent appropriate anticoagulation is recommended for acute complications, including LMW heparin, antithrombin III agonists such as Fondaparinux, or oral factor Xa inhibitors, such as apixaban or rivaroxaban [54,55] (26867832, 22315259). For more long-term management of COVID-19 inpatients that could not receive diagnostic imaging illustrating confirmation of thrombotic complications, continuation of anticoagulants or vitamin K antagonists, such as warfarin, are recommended for a short period of time after discharge [4]. Notably, current therapy of SARS-CoV-2 immunothrombosis is not currently directed at the cause of thrombotic complications, but rather a product of inflammation and endothelial damage. Treatment is currently focused on agonizing anti-coagulation factors like anti-thrombin III (i.e., heparin) or inhibiting coagulation factors through negative interference of vitamin K (i.e., warfarin) [56], but there appears to be little interest of considering inhibition of PAD or PAD4 isozyme to control citrullination and thereby NETosis-mediated thrombotic complications. The focus of the remainder of this review is to inform about the potential of PAD-4 inhibitors as therapeutic agents for clotting manifestations of SARS-CoV-2 infection.

## 5. PAD-4 Inhibitors as Therapeutic Agents and Potential as a COVID-19 Treatment

Administration of Cl-Amidine, a non-selective PAD4 inhibitor to mice with mastitis (inflammation of the breast tissue), is shown to reduce associated NET levels [57]. In mice induced with diabetes through injection of streptozotocin, associated with NETosis of wound locations, Cl-amidine has a similar effect of reducing NET levels and rescuing wound healing [58]. Further use of the inhibitor in mice where sepsis is induced shows a consistent rescue of originally elevated NET and citH3 levels [59]. Similar conclusions are apparent in numerous studies that use Cl-amidine as a way to reduce NET or citH3 levels in a variety of disease models, such as colitis, systemic lupus erythematosus, rheumatoid arthritis, and severe acute pancreatitis [60,61,62,63,64]. In in vitro analysis of COVID-19 patient neutrophils, demonstrated to produce increased NETs, incubation with Cl-amidine was shown to halt NET release, which correspondingly illustrated high numbers when inspecting lung tissues of post-mortem COVID-19 patients by microscopical analysis [11]. Additionally, SARS-CoV-2 activated neutrophils of the study were found to induce apoptosis of human alveolar basal epithelial cells and said apoptosis was subsequently inhibited by addition of Cl-amidine, suggesting that the use of Cl-amidine in potentially treating immunothrombosis may extend to alleviating lung tissue injury. A study observing the effects of a viral RNA analogue, polyinosinic-polycytidylic acid [poly(I:C)], discovered considerable results on the ability of the analogue to induce NET formation and citrullination of histones in a dose-dependent manner [44], consistent with byproducts of COVID-19 infection. Intriguingly, use of BB-Cl-amidine, a potent irreversible PAD-4 inhibitor, displayed complete attenuation of NET formation and stark suppression of citH3. Malignancies such as multiple myeloma, a malignancy of plasma cells overgrowth [65], have been shown to be positively responsive to administration of PAD-4 inhibitors. Multiple myeloma has been linked with upregulation of histone citrullination and NET formation, changes analogous to the inflammatory responses observed with SARS-CoV-2 infection [66]. PAD-4 inhibitor BMS-P5, a potent, selective, and relatively novel PAD4 inhibitor, is found to be just as effective in reducing NET levels and citrullinated histones, in comparison to less novel PAD-4 inhibitors such as Cl-amidine and GSK-484. In human neutrophils separated from patient sera and added to MM cell lines to induce NET formation, BMS-P5 provided more significant reduction of NETosis when compared to GSK-484, a reversible selective PAD4 inhibitor. Such results were consistent when examining mice infected with MM cell-lines that were subsequently provided with a PAD4 inhibitor, reducing symptoms consistent with multiple myeloma, as well as cell free-DNA and citrullinated histones, products associated with NET constituents and NET formation, respectively. This data not only demonstrates the possibility of PAD inhibitors to be used as therapeutic agents for COVID-19-mediated thrombosis, but also offers insight on PAD-4 inhibitor potency.

Despite PAD inhibitors providing a broad therapeutic coverage, ranging from inflammatory conditions to lung tissue injury, the mechanism of action of PAD inhibitors may suggest a potential for side effects, such as immunosuppression or undesirable cell death. Administration of Cl-amidine in LPS-induced mice mastitis models was observed to decrease the activation of transcription factor NF-κB signaling in addition to mRNA levels of TNF-α, IL-1β, and IL-6, beyond control expression [57]. These observations are concordant with a separate study that demonstrated that Cl-amidine use in TLR-agonized dendritic cell cultures significantly reduced TNF-α, IL-1β, IL-6, IL-10, and IL-12p70 [67]. NF-κB signaling has been demonstrated to play a crucial role in primary and secondary lymphoid development, B cell survival and maturation, and general innate and adaptive immune responses [68,69]. IL-6 has also been noted to engage B cell maturation, while IL-1β and TNF-α are involved with neutrophil mediation and T-cell activation, respectively [70,71]. Such marked attenuation of core immunoregulatory components suggests the potential of Cl-amidine to provide immunosuppressive effects, although literature suggests such suppressive effects may be dependent on cytokine and chemokine concentration. Cl-amidine administration in TLR-agonized dendritic cell cultures only attenuated cytokine production in the presence of increased concentrations, while baseline cytokine production was not affected [67]. One of the primary roles of NETs is to act as a nucleosome, cell, and protein-filled web that entraps bacteria, thereby yielding protective immunoregulatory effects. Presumably, PAD4 knockout mice were observed to have diminished NET formation and subsequently impaired survival rate from polymicrobial sepsis [58]. Remarkably, data analysis of Cl-amidine provided to LPS-induced endotoxic shock rabbits noted an increase in the TNF-α cytokine, rather the mRNA from TNF-α in an aforementioned study, compared to sham rabbits, insinuating that the pan-PAD inhibitor may not be involved in TNF-α attenuation specifically [72] and that immunosuppressive effects of Cl-amidine are potentially ambiguous. In a variety of different cancers, Cl-amidine yields cytotoxicity in human promyelocytic leukemia HL-60 and colon adenocarcinoma cell-lines [73]; however, this cytotoxic effect is not applicable to non-cancerous cell lines, including mouse fibroblast NIH-3T3 and HL-60 granulocytes. Cancer cell inhibition is also characteristic of more novel pan PAD inhibitors, such as YW3-56, demonstrated to suppress the growth of triple negative breast cancer cells with absence of injury toward non-cancerogenic MCF10A breast epithelial cells [74]. The prior data presented on the immunosuppressive and cancer regulatory abilities of PAD inhibitors proposes that PAD inhibitors could be dosed within therapeutic windows to minimize side effects of immune system depression and wild type cell destruction.

## 6. PAD-4 Inhibitors Attenuate Immunothrombosis and Acute Lung Injury Complications

It was demonstrated that increased NETs and citH3 levels are correlated to states of immunothrombosis in both animal and human models [5,23]. A variety of PAD4 inhibitors have been discussed, displaying the ability of inhibition to reduce NET and citH3 levels [56,75], leading one to hypothesize a potential reduction in immunothrombosis (Figure 1). A study that attempts to close this gap of curiosity utilizes Cl-amidine in mice with induced middle cerebral arterial occlusion through a modified photothrombotic method [75]. Fascinatingly, administration of the PAD inhibitor before induction of arterial occlusion resulted in a lack of thrombus formation, indicating that NETs trigger thrombosis through citrullinated histones. Mice induced with arterial occlusion by use of ferric chloride (FeCl_3_) or wire injury were concordantly found to have lower NETs and a reduction of arterial thrombi when exposed to Cl-Amidine [56]. In addition to reducing arterial thrombi, Cl-amidine shows potential to minimize atherosclerotic thrombi in mice models, displaying a dynamicity in the ability of the inhibitor to be effective in decreasing the surface area of thrombi of various constituents [76]. Cl-amidine use appears effective across a variety of pathologies, extending to autoimmune disorders, such as systemic lupus erythematosus (SLE), characteristic of premature vascular disease [62]. Incubation of sections of SLE-model mice aorta with PMA stimulated neutrophils to produce NETs and the addition of Cl-amidine lead to the observation of fewer NETs and less endothelial damage to the aorta sections than mice without the PAD inhibitor. Interestingly, use of the inhibitor in the SLE-model mice significantly improved the ability of bone marrow progenitor cells to differentiate into mature endothelial cells and improved endothelial-dependent vasorelaxation of aortas, consistent with a similar study by the respective author [77]. Use of Cl-amidine in SLE-model mice did not produce anti-thrombotic results as strongly as noted in Martínez et al., showing a significant prolongation of thrombus formation rather than a complete abrogation of thrombus formation.

Specific PAD4 inhibitors, such as GSK-484, appear to show equivalent efficacy to non-specific PAD inhibitors such as Cl-amidine. Heparin-induced thrombocytopenia (HIT) is a life-threatening immune reaction to heparin administration that can ultimately result in NETosis and thrombi containing NETs [47]. Providing GSK-484 to HIT-induced mice resulted in complete abrogation of NET-infiltrated thrombus formation and expectedly obliterated citH3 in the plasma of mice. When observing a different thrombotic-inducing pathology, myocardial infarction in mice, mice that underwent left coronary artery ligation to mimic a myocardial infarction, were found to have significantly reduced thrombi size and a correspondingly decreased citH3 [78]. Noting that a majority of pathology discussed is not directly translatable to pathology observed in SARS-CoV-2 infection, acute lung injury in mice has been found to be improved through administration of a selective PAD-4 inhibitor [50]. Similar to GSK-484, Thr-Asp-F-amidine (TDFA) is PAD-4 inhibitor, specifically a tripeptide. In mice subject to intratracheal lipopolysaccharide to induce acute lung injury, administration of TDFA prior to induction resulted in an inspiring decrease in lung injury as well as a significant decrease in lung histopathology, strikingly similar to the control mice. 2-chloroacetamidine, a relatively novel covalent PAD-4 inhibitor [24], was found to be helpful in reducing hypercitrullination and inflammation in mice that were airway challenged to mimic an environment of allergic airway inflammation [79]. Aforementioned, use of viral RNA analogue [poly(I:C)] to induce NETosis and hypercitrullination and subsequent administration of BB-Cl-Amidine demonstrated a marked decrease of both products [44]. Use of [poly(I:C)] surprisingly mimicked SARS-CoV-2 thrombotic complications, resulting in an increase in thrombus formation of post-mortem mice lung tissues. Dispensation of PAD-4 inhibitor, BB-Cl-Amidine, not only reduced the potential markers of thrombosis but further obliterated thrombus formation, suggesting that the PAD-4 inhibitor has a protective effect against immunothrombosis. Furthermore, use of BB-Cl-Amidine abrogated the [poly(I:C)] response of neutrophil infiltration, fibrin deposition, disruption of barrier-permeability function of lung vascular endothelial cells, and interestingly reversed thrombocytopenia. Such observations increase the likelihood of implications of PAD inhibitors being used to treat thrombosis more easily applicable to the specific pathology of SARS-CoV-2 and thrombotic complications that can arise from acute lung injury.

## 7. Conclusions

As of 16 August 2021, some PAD-4 inhibitors such as taxol, streptomycin, and minocycline have been available for the treatment of various diseases. Streptomycin is traditionally used as an antibiotic for the treatment of infections such as tuberculosis [80], minocycline for the treatment of dermatologic conditions such as acne vulgaris [81], and taxol for various cancers. Importantly, these inhibitors are weak and non-specific PAD-4 inhibitors that have not yet been pre-clinically tested in non-human animal models with SARS-CoV-2 and appear inferior to the more potent PAD-4 inhibitors such as Cl-Amidine, GSK 484, BB-Cl-Amidine, and TDFA. Opportunistically, NET-inhibitor factor and related peptides present in human umbilical cord blood have been found to negatively modulate NET activity and PAD-4 activity parallel to Cl-amidine [82], evidenced to be a potent PAD-4 inhibitor [24]. The advent of this endogenous peptide is undergoing continuous research and serves as a novel anti-inflammatory and anti-thrombotic treatment to viral infections, such as SARS-CoV-2 [16,81,82,83,84]. Principally, existing literature has not tested the efficacy of PAD-4 inhibitors in inflammatory conditions, such as COVID-19 in human models. With the continuous discovery of SARS-CoV-2 variants, including but not limited to B.1.1.7 (Alpha), B.1.351 (Beta), P.1 (Gamma), B.1.617.1 (Kappa) B.1.617.2 (Delta), B.1.526 (Iota), B.1.351, A.23.1, and most recently, the highly convoluted B.1.1.529 (Omicron) (Callaway, 2021) [85,86,87,88], SARS-CoV-2 treatment and vaccination are both constantly challenged. Although specific mutations in the spike protein, including the B.1.1.7 and B.1.351 variants, are generally effectively neutralized by existing vaccines [85], the response of vaccines to other variants, including the P.1 andB.1.617.2 display significantly reduced neutralization of SARS-CoV-2, with other variants such as the B.1.1.529 posing as novel challenges toward herd immunity. The B.1.617.2 variant, initially identified in India [89], possesses mutations granting enhanced transmissibility and producing more severe symptoms characteristic of higher viral loads within the respiratory tract, in addition to a unique symptom of gangrene secondary to severe blood clots. The anti-inflammatory effects of PAD inhibitors in combination with immunothrombosis suppression through NET inhibition offer a potential acute treatment for severe SARS-CoV-2 patients in whom vaccine protection is breached, in the case of more resistant strains. Noting that existing literature has exclusively tested PAD-inhibitors in non-human animal models, generalizability of inhibitory responses is difficult to interpret in human models; however, prior studies have identified minimal side effects in administering PAD inhibitors to mice, rabbits, or rhesus monkeys. Ultimately, ambiguity of the efficacy and side effects of these inhibitors in humans requires continuing research to unveil the potential of PAD-4 inhibition in SARS-CoV-2 immunothrombosis.

## Figures and Tables

**Figure 1 biomedicines-09-01867-f001:**
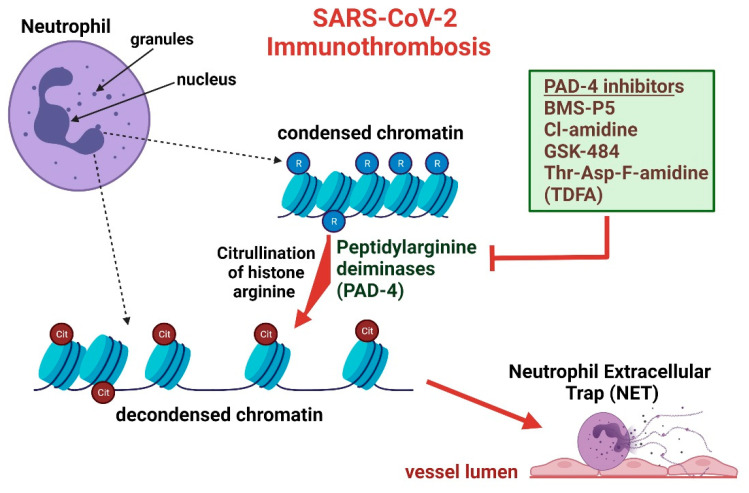
SARS-CoV-2 infection and subsequent inflammation alters neutrophil chromatin, ultimately producing neutrophil extracellular traps that are vulnerable to attenuation through PAD-4 inhibition. Neutrophil chromatin is non-pathologically found in a condensed state mediated through arginine (R) residues. In an inflammatory state, condensed chromatin is modified to a decondensed form through histone citrullination (Cit) by peptidylarginine deiminases (PAD-4), thereby neutralizing DNA-backbone interactions. Decondensed chromatin and various proteins not displayed in the image, produce “net-like” occlusions that mount in the vessel lumen, resulting in thrombus formation. Use of various PAD-4 inhibitors cease the formation of decondensed chromatin at varying affinities, a necessary chromatin modification for NETs formation.

**Table 1 biomedicines-09-01867-t001:** From left to right, the table illustrates a specific PAD inhibitor, along with respective IC_50_ and K_i_ constant, and additionally displays PAD isozymes 1–4. Decreased IC_50_ values represent increased potency by requiring less inhibitor to produce half of a desired effect, while the inhibitory constant further demonstrates potency through analysis of the concentration required to produce half of the maximum inhibition. Notably, GSK484 and GSK 199 IC_5o_ were collected in the absence of Ca^2+,^ being that the ion acts as a competitive inhibitor for enzyme-substrate activity. ND represents “Not Determined”, which is applicable to more novel or less researched PAD inhibitors. PAD-selective IC_50_ or K_i_ information may be excluded from the table if isozyme specificity is unknown.

PAD Inhibitor	PAD1	PAD2	PAD3	PAD4
Cl-Amidine				
IC50 (µM)	0.8 ± 0.3 µM	17 ± 3.1 µM	6.2 ± 1.0 µM	5.9 ± 0.3 µM
K_I_ µM	62 ± 11 µM	ND	28 ± 7.3 µM	180 ± 33 µM
TDFA				
IC50 (µM)	8.5 ± 0.8 µM	71 ± 4.4 µM	26 ± 7.4 µM	2.3 ± 0.2 µM
K_I_ (µM)	ND	ND	180 ± 60 µM	16 ± 10 µM
F-Amidine				
IC50 (µM)	30 ± 1.3 µM	51 ± 9.0 µM	≥350 µM	22 ± 2.1 µM
K_I_ (µM)	110 ± 40 µM	ND	290 ± 190 µM	330 ± 90 µM
YW3-56				
IC50 (µM)	ND	0.5–1 µM	ND	1–5 µM
K_I_ (µM)	ND	ND	ND	ND
GSK484				
IC50 (µM)	ND	ND	ND	0.05 µM
KI (µM)	ND	ND	ND	ND
GSK199				
IC50 (µM)	ND	ND	ND	
KI (µM)	ND	ND	ND	0.2 µM
Paclitaxel (Taxol)				
IC50 (µM)	ND	ND	ND	5 × 10^3^ µM
KI (µM)	ND	ND	ND	ND

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
