# Peer review of "PAD Inhibitors as a Potential Treatment for SARS-CoV-2 Immunothrombosis"

_biomedicines, 2021, doi:10.3390/biomedicines9121867_

Round 1

Reviewer 1 Report

Coronavirus disease 2019 (COVID-19) is an infectious disease caused by severe acute respiratory syndrome coronavirus-2. Immunothrombosis significantly contributes to the pathophysiology of COVID-19. The current submitted manuscript summarized current treatment of SARS-CoV-2 immunothrombosis and reviewed the PADs in SARS-CoV-2 and animal experimental results about PAD-4 Inhibitors in PADs Covid-19 treatment. They concluded that PAD inhibitors, especially PAD-4 Inhibitors may serves as a novel anti-inflammatory and anti-thrombotic treatment to viral infections such as SARS-CoV-2. This topic is interesting for a board readership, especially for scientists dealing with immunothrombosis and viral infections. I have only one suggestions before this review is suitable for publication.  The side effects of these inhibitors, as authors mentioned in last of MS, may need continuing research. However, the side effects of inhibitors in animal experiments or in other clinical application should be discussed.  

Author Response

Reviewer one addressed the lack of content dedicated toward the side effects of PAD inhibitors tested on animals or in other clinical applications; this concern was addressed throughout lines 291-326. It was noted that not much literature is present directly studying the side effects of PAD4 inhibitors. This information on PAD side effects focuses on potential immunosuppressive effects and unwanted cell death from PAD cytotoxic effects on cancer cells. Lines 305-308 suggest that PAD immunosuppression is dependent on inflammation, while lines 309-312 notes that attenuating NETs through PAD inhibition can affect microbial clearance, increasing the likelihood of polymicrobial sepsis. PAD inhibitors are also shown not to affect healthy cell lines when used for cancer cell destruction, stated in line 319.

Reviewer 2 Report

Minor revision

The authors review clinical results of PAD-4 inhibitors in reducing thrombotic complications across various inflammatory disorders like Covid-19 and suggested that these inhibitors may be valuable in treating the origin of SARS-CoV-2 immunothrombosis. The work is well written and the topic is of importance, though I have a few comments and hopefully the authors can reflect on them and improve the manuscript.

  1. There have been a few papers reporting PAD Inhibitors, the authors should provide a table to clearly present, compare, and analyze their results in order to distill useful info for the field.
  2. As a review paper, the authors should expand content in Perspective. Such as discussing the potential implications of PAD inhibitors for treating new Covid variants.

Author Response

 The reviewer had two comments towards the paper: providing a table summarizing useful information of PAD inhibitors and expanding content in perspective. In respect to the first comment, a table was inserted after line 214, following a paragraph introducing the various PAD inhibitors and their potencies, measured by IC50 and KI. The table summarizes the various PAD inhibitor’s IC50 and KI values toward different isozymes of PAD. Noting the final comment on expanding the content within the perspective, lines 412-428 ultimately focuses on informing the scientific community of Covid-19 variants and their challenge toward treatment and vaccination, as well as the ability to use PAD inhibitors to treat more severe symptoms, such as moderate to severe inflammation and thrombosis.